# Learning to Elect

**Cem Anil**[*]
University of Toronto
Vector Institute
anilcem@cs.toronto.edu

**Xuchan Bao**[*]
University of Toronto
Vector Institute
jennybao@cs.toronto.edu

## Abstract

Voting systems have a wide range of applications including recommender systems, web search, product design and elections. Limited by the lack of general-purpose analytical tools, it is difficult to hand-engineer desirable voting rules for each use case. For this reason, it is appealing to automatically discover voting rules geared towards each scenario. In this paper, we show that set-input neural network architectures such as Set Transformers, fully-connected graph networks and DeepSets are both theoretically and empirically well-suited for learning voting rules. In particular, we show that these network models can not only mimic a number of existing voting rules to compelling accuracy — both position-based (such as Plurality and Borda) and comparison-based (such as Kemeny, Copeland and Maximin) — but also discover near-optimal voting rules that maximize different social welfare functions. Furthermore, the learned voting rules generalize well to different voter utility distributions and election sizes unseen during training.

## 1 Introduction

Voting systems are highly prevalent in our daily lives. Examples range from large scale democratic elections to company or family-wide decision making, recommender systems and product design [Boutilier et al., 2015].

As with any social decision-making process, the goal of designing voting rules is to reconcile differences and maximize some collective objective. The area of research that studies different voting rules and the approaches to designing them is called voting theory.

A vast number of voting rules have been proposed over the years. Among them is the widely applied plurality rule. Despite being simple and intuitive, the plurality rule is very limited in that it does not consider the strength of voters' preferences. Other examples of voting rules, such as Borda and Copeland, take into consideration the ranked preferences of the voters.

Voting theorists have developed different approaches to designing voting rules. For example, the axiomatic approach constrains the voting rules to satisfy certain desired properties (axioms) such as anonymity (treating all voters equally) and neutrality (treating all candidates equally). The utilitarian approach, on the other hand, aims to maximize a pre-defined notion of social welfare — a scalar quantity that measures the quality of the elected candidate in the eyes of the electorate.

There are major hurdles to overcome in the traditional way of designing and implementing voting rules. First, the celebrated Arrow's Theorem states the nonexistence of non-dictatorship voting rules that simultaneously satisfy a set of seemingly sensible axioms [Arrow et al., 1951]. Second, for some voting rules such as the ones based on pairwise comparisons, finding the winner can be computationally expensive, making them infeasible for large-scale applications. Last but not least, for the utilitarian approach, it is not obvious how to design voting rules that maximize a given notion

---

[*]Equal contribution.

35th Conference on Neural Information Processing Systems (NeurIPS 2021).

of social welfare.[2] There might be a diverse set of social welfare functions of interest, but theory is lacking in finding their corresponding optimal voting rules.

In this paper, we aim to tackle the latter two limitations using neural networks. While doing so, we also seek to preserve certain desired properties such as voter anonymity. In particular, we identify three permutation-invariant neural network (PIN) architectures: DeepSet [Zaheer et al., 2017], Set Transformer [Lee et al., 2019], and Graph Isomorphism Network (GIN) [Xu et al., 2018], and apply them to learn the voting rules. As compact and universal function approximators, such trained neural networks not only address the computational burden of some voting rules, but also provide a flexible approach to maximize a diverse set of social welfare functions.

The main contributions of this paper include:

- We show that PIN architectures are theoretically and empirically well-suited for learning to represent voting rules. Theoretically, we show that the three PIN architectures are universal approximators in the space of permutation-invariant functions. This includes a novel proof on the universality of Graph Isomorphism Networks with fully-connected graphs (as in learning voting rules).
- We apply the aforementioned PIN models to mimic five classical voting rules: plurality, Borda, Copeland, Maximin and Kemeny. We show that they can mimic these voting rules to compelling accuracy and can generalize seamlessly to unseen real datasets and elections with an unseen number of voters.
- We train the networks to maximize two different social welfare functions — utilitarian and egalitarian — on elections sampled using three different underlying voter utility distributions. We show that the PIN models *discover novel voting rules* that maximize social welfare better than classical voting rules. In cases where theoretical optimal voting rules are known (i.e. for the utilitarian social welfare function), the PIN models match the optimal performance.

The organization of this paper is as follows. Section 2 provides background on voting theory. Section 3 describes our method of using PINs to learn voting rules. Specifically, we introduce the permutation-invariant network architectures (Section 3.1), show the universality results (Section 3.2), and describe the proposed the training procedure for learning voting rules in detail. In Section 4, we show comprehensive experiment results on the effectiveness of PIN models in learning voting rules. We discuss related works in Section 5, limitations in Section 6 and conclude with Section 7.

## 2  Background

### 2.1  Voting theory preliminaries

We adopt the formalism used by Boutilier et al. [2015]. Let $N = \{1, \ldots, n\}$ be a set of voters, $A = \{a_1, \ldots, a_m\}$ be a set of candidates and $R = \{1, \ldots, m\}$ be the integer rankings each voter can assign to a candidate. Each vote $i$ is represented as a bijection from candidates to rankings: $\sigma_i : A \mapsto R$. The vector of candidate preferences $\vec{\sigma} = (\sigma_i, \ldots, \sigma_n) \in (S_m)^n$ is called a preference profile. A voting rule $f : (S_m)^n \mapsto A$ maps preference profiles to candidates.

The utilitarian approach to voting makes the assumption that voters have *utility functions* ($u : A \mapsto \mathbb{R}_+$) over the alternatives which quantify how much a voter prefers a candidate. Utility functions are consistent with the rankings — that is, candidates with higher utilities are ranked higher by voters. The vector of all voters' utility functions $\vec{u} = (u_1, \ldots, u_n)$ is called a *utility profile*. The social welfare of an alternative $sw(a, \vec{u}) : A \times \mathbb{R}^{n \times m} \mapsto \mathbb{R}$ quantifies the "desirability" of a candidate $a$ under a given preference profile $\vec{u}$. The utilitarian approach posits that the ultimate goal of a voting rule is to pick the candidate that maximizes social welfare. Note that since voting rules only have access to rankings and not the utilities, this is often impossible to achieve without further assumptions.

A popular notion of social welfare is the utilitarian social welfare function, which computes the sum of all the utilities the voters assign a candidate $sw_{\text{util}}(a, \vec{u}) = \sum_{i \in N} u_i(a)$. There exists, however, many different social welfare functions. For example, the Rawlsian social welfare function aims to make even the least happy voter as happy as possible: $sw_{\text{rawl}}(a, \vec{u}) = \min_i u_i(a)$, and the egalitarian social

---

[2]We consider average-case social welfare maximization, unlike the worst-case analysis in the distortion literature [Boutilier et al., 2015, Caragiannis et al., 2017].

welfare function aims to maximize the utilitarian welfare, regularized by a penalty term promoting equality: $sw_{\text{egal}}(a, \vec{u}) = \sum_{i \in N} u_i(a) - \lambda \sum_{i \in N}(u_i(a) - \min_j(u_j(a)))$ [Allcott, 2011]. It is up to the designer of the voting system to pick a notion of social welfare best suited for the task at hand.

## 2.2 Classical voting rules

Most classical voting rules can be classified under two groups: score based rules and comparison based rules. Score based rules (also called scoring functions) are defined by a score vector $\vec{s} = (s_1 \ldots s_m)$. Each candidate $a \in A$ is assigned the score $\sum_{i \in N} s_{\sigma_i(a)}$ and the candidate with the largest score is picked as the winner [Boutilier et al., 2015]. Famous examples include Plurality (with a score vector of $\vec{s}_{\text{plr}} = (1, 0, \ldots, 0)$) and Borda (with a score vector of $\vec{s}_{\text{borda}} = (m - 1, m - 2, \ldots, 0)$). Comparison based rules operate using pairwise comparison matrices $R \in \mathbb{R}^{m \times m}$ whose entries are filled based on how candidates fare against each other in pairwise comparisons. Well-known examples include the Copeland, Maximin and Kemeny rules. The Copeland rule picks the candidate who fares better in pairwise comparisons the largest number of times: $\arg\max_i(\sum_j r_{ij})$ where $r_{ij}$ stands for the number of times candidate $i$ fares better against candidate $j$ in the voter preferences. Maximin, also known as Simpson's rule, picks the candidate for whom the candidate who fares the best against him/her in pairwise comparisons has the least pairwise score: $\arg\min_i(\max_j r_{ji})$ [Bubboloni et al., 2020]. The Kemeny rule [Kemeny, 1959] first computes a ranking that maximizes the sum of all pairwise wins: $\sigma^* = \arg\max_\sigma \sum_{i \succ_\sigma j} r_{ij}$, where $i \succ_\sigma j$ means that candidate $i$ is preferred against $j$ according to ranking $\sigma$. The Kemeny winner is the candidate that ranks the first in $\sigma^*$. Note that computing the Kemeny ranking or Kemeny winner is NP-hard [Bartholdi et al., 1989].

## 2.3 Average-case optimal voting rules

In cases where the social welfare function takes a very simple algebraic form, it might be possible to derive the voting rule that achieves the largest expected social welfare under a given utility distribution $P_u$. For example, it is known that the average-case optimal voting rule for the utilitarian social welfare funtion is a score-based voting rule with the score vectors $s_k^* = \mathbb{E}_{u \sim P_u}[u(a)|(\sigma(u))(a) = k]$ (or, the average utility of the candidates that are ranked at $k^{\text{th}}$ position) [Boutilier et al., 2015].

## 3 Permutation-Invariant Networks (PINs) to learn set-input functions

We review the fundamentals of learning set-input functions through the use of permutation-invariant networks (PINs). Learning set-input functions is desirable in the context of learning voting rules: 1) it enables processing elections of varying sizes (i.e. different number of voters) 2) it makes it possible to satisfy desirable properties such as anonymity (invariance to the shuffling of voter identities) through architectural constraints. We review three permutation-invariant architectures (DeepSets, fully-connected graph networks and Set Transformers) and show that they're universal approximators of set-input functions. We also detail the training procedure for learning voting rules.

### 3.1 Constructing permutation-invariant architectures

Functions defined on sets are by definition *permutation-invariant* [Zaheer et al., 2017].

**Property 1.** *Let $2^{\mathcal{X}}$ represent the powerset of $\mathcal{X}$. Any set-input function $f : 2^{\mathcal{X}} \mapsto \mathcal{Y}$ must be invariant to the reordering of its inputs by any permutation $\sigma$: $f(\{x_1, \ldots, x_M\}) = f(\{x_{\sigma(1)}, \ldots, x_{\sigma(M)}\})$*

Zaheer et al. [2017] showed that one can practically train expressive set-input neural networks by chaining together a permutation-equivariant feature extractor (encoder) and a permutation-invariant decoder.[3] permutation-equivariance is the property that when the inputs of a function are permuted, the outputs get permuted the same way:

**Property 2.** *A function $f : \mathbb{R}^{M \times d_{in}} \mapsto \mathbb{R}^{M \times d_{out}}$ is permutation-equivariant if $f([x_{\sigma(1)}, \ldots, x_{\sigma(M)}]) = [f_{\sigma(1)}(\mathbf{x}), \ldots, f_{\sigma(M)}(\mathbf{x})]$*

---

[3]Composition of permutation-equivariant and permutation-invariant functions results in permutation-invariant functions.

All of the architectures described below follow the central design principle of composing permutation-equivariant building blocks with permutation-invariant ones to build expressive set-input networks.

**DeepSets [Zaheer et al., 2017]** This architecture first encodes each element of the set independently using an encoder network (e.g. a multilayer perceptron), pools the outputs of the encoder (e.g. using sum, max or mean pooling) and finally passes the result through a decoder network (e.g. another multilayer perceptron). Since the encoder treats the elements of the set independently to achieve permutation-equivariance, the pooling operation is the only step that can model interactions between the elements in the set.

**(Fully-connected) graph neural networks (GNNs)** Instead of achieving permutation-equivariance through the independent processing of the set elements, we can instead view the input set as a *fully-connected graph* and build a GNN-based [Scarselli et al., 2008] encoder that takes into account *all* of the interactions between different elements of the set. Each GNN layer transforms the nodes in the graph by concatenating a permutation-invariant aggregation of features of the neighbour nodes and a transformation that produces the next-layer features given the current-layer features and the result of aggregation. A GNN consists of multiple GNN layers, and the graph-level embedding is obtained by *pooling* all the node features using a permutation-invariant function. In our experiments, we use a powerful variant of GNN — the Graph Isomorphism Network (GIN) [Xu et al., 2018]. Detailed background on GNNs and GIN can be found in Supplementary Material.

For a fully-connected graph, the graph structure remains unchanged under node permutation. Since both the neighbourhood aggregation and the graph-level pooling operations are permutation-invariant, the whole GNN represents a permutation-invariant function.

**Set Transformers [Lee et al., 2019]** The computational building block of set-transformers is the Query-Key-Value (QKV) attention [Vaswani et al., 2017], which can be interpreted as a differentiable dictionary retrieval operation. The neural network controls both the input and output space of the QKV operation using *multihead attention*, which multiplies all the input and output tensors by learnable weight matrices.

The multihead attention possess desirable permutation-equivariance/invariance properties for learning set-input functions. Set Transformer builds highly expressive encoder-decoder architecture with multiple multihead attention blocks, resulting in the overall permutation-invariance [Lee et al., 2019]. Detailed background and the full encoder-decoder architecture are found in Supplementary Material.

### 3.2 Universality results

An important property that we would like these neural network architectures to possess is universal approximation in the space of permutation-invariant functions. Since any anonymous voting rule can be expressed as a permutation-invariant function without the presence of ties, a universal approximating network has the maximum representational power over these voting rules.

Throughout this section, we assume that the input feature space $\mathcal{X}$ is countable. We also assume that the output space is $\mathbb{R}$. These assumptions are appropriate in the case of approximating voting rules.[4]

The universal approximating properties of DeepSets and the Set Transformer have been established [Zaheer et al., 2017, Lee et al., 2019]. We establish the universal approximating results of GIN in the following subsection.

#### 3.2.1 Universality of GIN

Xu et al. [2018] showed that the representational power of GIN is equal to the power of the Weisfeiler-Lehman (WL) graph isomorphism test [Weisfeiler and Leman, 1968]. We extend their result to show that in the case of fully-connected graph structures (as in learning voting rules), GIN has equal representational power as the full graph isomorphism testing. We state this result formally below:

**Theorem 1.** *Let $G_1, G_2 \in \mathcal{G}$ be any two non-isomorphic fully-connected coloured graphs. Let $\mathcal{A} : \mathcal{G} \to \mathbb{R}^d$ be a GNN that maps any two graphs that the Weisfeiler-Lehman test of isomorphism decides as non-isomorphic to different embeddings in $\mathbb{R}^d$. Then $\mathcal{A}$ maps $G_1$ and $G_2$ to different embeddings.*

---

[4]Even though the networks output logits in $\mathbb{R}^m$, it is sufficient to show the networks are universal in representing a scalar function with a maximum on the logits.

---

**Algorithm 1:** Supervised Learning of Voting Rules

---

**Inputs** : Utility distribution $P_u$,
                    Existing voting rule or oracle $f$
Initialize the weights of model $f_{\mathcal{W}}$
**while** *loss not converged* **do**
    Sample voter utilities: $\vec{u} \sim P_u$
    Compute preference profiles: $\vec{\sigma}(\vec{u})$
    Get target winner $y = f(\vec{\sigma})$ (existing rule)
             or $y = f(\vec{u})$ (oracle)
    Get predicted winner $\hat{y} = f_{\mathcal{W}}(\vec{\sigma})$
    Compute prediction loss: $l = l(\hat{y}, y)$
    Compute gradients $\frac{\partial l}{\partial \mathcal{W}}$ and update weights $\mathcal{W}$
**end**
**return** Learned rule $f_{\mathcal{W}}$

---

The proof hinges upon the observation that when applied to fully-connected coloured graphs, the Weisfeiler-Lehman iterations do not alter the colouring of nodes. We defer the formal proof to Supplementary Material.

In order to establish universality results from Theorem 1, we refer to Chen et al. [2019], who established the equivalence between graph isomorphism and the universal approximation of permutation-invariant functions. In particular, Chen et al. [2019] proved that if a GNN is graph isomorphism-discriminating, then the GNN with additional two feed-forward layers is universal approximating.

## 3.3 Learning voting rules

We would like to train neural networks to represent voting rules — the network takes in the voter preference profile, and outputs the predicted winner.[5] In this paper, we are interested in training networks to perform two tasks: 1) to mimic existing voting rules, and 2) to discover novel voting rules that maximize some notion of social welfare. The training data for the first task naturally provides input-label pairs, making it suitable for supervised learning. For the second task, different learning approaches are possible.

**Maximizing social welfare** Under the utilitarian framework, the social welfare is a function of the underlying utilities that voters assign to the candidates. In reality, neither the voter utilities nor the social welfare quantity are easily accessible. However, if we make assumptions on the *distribution* from which the voter utilities are generated, we can generate synthetic training data and get access to the utility information.

It is possible to frame the social welfare maximization as a reinforcement learning (RL) problem, or more specifically, as a contextual bandits problem. The network would learn to maximize the reward (social welfare) by choosing an action (the winner candidate) from a discrete set, given some situation or context (the voter preference profile).

A more straightforward alternative is to use supervised learning. Since we have access to the utilities at training time, we can define an *oracle* "voting rule" that takes in the utilities and outputs the winner that maximizes social welfare.[6] We can then train the networks using supervised learning, with the oracle output as the target. This approach is also called *behaviour cloning*. Compared to RL, it has the weakness of penalizing all wrong predictions equally. However, it is easier to optimize and empirically yields strong results (Section 4). We leave the RL approach to future work.

Since the training procedure is identical to that of the first task (except that we mimic the oracle instead of an existing voting rule), we summarize the training procedure for both tasks in Algorithm 1.

---

[5]For detailed discussion about the input and output representations, see Supplementary Material.
[6]The oracle is not an actual voting rule, as it uses information (utilities) that is not accessible to voting rules.

| Architect. | Mimicking Accuracy | | | | |
| --- | --- | --- | --- | --- | --- |
| | Plurality | Borda | Copeland | Maximin | Kemeny |
| Set Trans. | **1.0** | **0.99** | 0.82 | **0.80** | **0.94** |
| GIN | **1.0** | **0.99** | 0.81 | 0.77 | 0.82 |
| DeepSets | **1.0** | 0.96 | **0.83** | 0.78 | 0.89 |
| MLP | **1.0** | 0.95 | 0.81 | 0.75 | 0.76 |

Table 1: Voting rule mimicking accuracy of learned voting rules. The entries represent the proportion of times the learned voting rule successfully predicts the output of the Plurality, Borda, Copeland, Maximin and Kemeny rules (higher is better). Permutation-invariant architectures achieve near perfect accuracy in approximating score-based voting rules (Plurality and Borda) and compelling accuracy in comparison-based rules (Copeland, Maximin and Kemeny).

## 4 Experiments

In this section, we investigate the effectiveness of PIN models on 1) mimicking classical voting rules (Section 4.1) and 2) discovering novel voting rules that maximize social welfare (Section 4.2).

**Training data generation** We used synthetically generated elections to train the networks. In our experiments, we sampled elections with numbers of voters and candidates ranging between 2 to 99 and 2 to 29 respectively (except for mimicking the Kemeny rule, where we sampled 3 to 10 candidates due to solver and computation constraints). We sampled the (normalized) voter utilities from the Dirichlet distribution with concentration parameters $\alpha \in \mathbb{R}^{n_c}$, where $n_c$ is the number of candidates. Sampling from the Dirichlet distribution ensures that the sampled utilities are non-negative and sum up to one. Different Dirichlet parameters lead to qualitatively different elections. We consider the following cases where the utility distribution is symmetric to each candidate, i.e. $\alpha = [\alpha_0 \ldots \alpha_0]$.[7] Specifically, we used $\alpha_0 = 0.5$ and 2 for "polarized" and "indecisive" in our experiments, respectively.

- "Polarized" ($0 < \alpha_0 < 1$): each voter likely strongly prefers a candidate.
- "Uniform" ($\alpha_0 = 1$): all utility profiles are assigned equal probability.
- "Indecisive" ($\alpha_0 > 1$): voters likely have no strong preference for the candidates.

We then used the generated utilities to compute the voter preference profiles. We also computed the target winner either based on an existing voting rule[8] (Section 4.1) or an oracle that maximizes social welfare based on the utilities (Section 4.2). The neural networks attempt to predict the winner using *only the voter preference profiles* (pre-processed with the one-hot candidate id function described in Supplementary Material).

To evaluate the diversity of the synthetically generated data, we empirically estimated the probability of pairs of traditional voting rules producing the same winner, using the "uniform" utility distribution (Table 5 in Supplementary Material). We conclude that the synthetically generated elections are diverse, hence can differentiate the voting rules.

**Network architecture details** We compared the performance of DeepSets [Zaheer et al., 2017], Graph Isomorphism Networks [Xu et al., 2018] and Set Transformers [Lee et al., 2019]. To make the results as comparable as possible, we constructed all architectures to have roughly 10 million parameters and comparable depth. We also used the same normalization layer (LayerNorm [Ba et al., 2016]) in all of the architectures. We additionally trained similar-sized multilayer perceptrons (MLP) as baseline. Further details of the architectures are described in Supplementary Material.

**Training setup** We used the Lookahead optimizer [Zhang et al., 2019] to train the DeepSet models, and the Adam optimizer [Kingma and Ba, 2015] to train the other networks. We tuned the learning rate for each architecture. We used the cosine learning rate decay schedule with 160 steps of linear warmup period [Goyal et al., 2017]. We used a sample size of 64 elections per gradient step. We trained each PIN model for 320,000 gradient steps. We trained each MLP model for three times as long (960,000 gradient steps), as MLP models are observed to learn more slowly. Additional details for the training setup are included in the Supplementary Material.

---

[7]The outcomes of these elections are difficult to predict, as no candidate is obviously preferred.

[8]For the mimicking tasks, we removed elections that have tied winners except for the Kemeny rule, in which case the integer programming solver we used does not efficiently detect the existence of ties.

## 4.1 Mimicking classical voting rules

Are Permutation-Invariant Networks (PIN) expressive enough to represent a number of classical voting rules? How does their performance vary as we move from computationally cheap position-based voting rules, such as Plurality and Borda, to more computationally demanding comparison-based ones, such as Copeland, Maximin and Kemeny? We generated synthetic data with the "uniform" utility distribution, and trained PINs to mimic these aforementioned voting rules. We then tested the robustness of the learned voting rules by computing how their mimicking performance varies on elections with different number of voters and candidates, elections from different data distributions and elections sampled from real world datasets.

**Mimicking accuracy** The performance of all architectures on mimicking the Plurality, Borda, Copeland, Maximin and Kemeny rules are presented in Table 1. For each value in Table 1, we sample 16,384 elections from the training distribution. A first observation is that PINs achieve near perfect accuracy in approximating score-based voting rules. This is a significant result, as many theoretically optimal voting rules are known to be score-based [Boutilier et al., 2015, Young, 1975]. Secondly, PINs outperform MLP models in all cases (except for Plurality, which all models

| Architect. | Mimicking Accuracy for Different Number of Voters | | | |
| --- | --- | --- | --- | --- |
| | (within-domain) | | (out-of-domain) | |
| | 2-49 | 50-99 | 100-149 | 150-199 |
| Set Trans. | **0.99** | **0.99** | **0.99** | **0.99** |
| GIN | **0.99** | **0.99** | 0.98 | 0.98 |
| DeepSets | 0.97 | 0.96 | 0.95 | 0.95 |
| MLP | 0.96 | 0.93 | N/A | N/A |

Table 2: The accuracy of permutation-invariant networks on predicting the Borda winner (higher is better). Even though the networks were trained on elections with less than 100 voters, the learned rules generalize seamlessly to unseen numbers of voters much larger than encountered during training.

learned perfectly). We would like to highlight that PINs achieve high accuracy in mimicking the Kemeny rule (significantly better than the MLP baseline), which is NP-hard to compute exactly. This shows that PINs have high potential as efficient approximate solutions for computationally expensive voting rules.[9]

**Generalization to unseen numbers of voters** Can PINs generalize to elections with a number of voters bigger than that encountered during training? We tested the PIN models on elections with voters between 2 and 199 (the models are trained on maximum 99 voters). We report the results in mimicking Borda in Table 2. Similar results for mimicking Plurality, Copeland, Maximin and Kemeny can be found in Supplementary Material. This confirms that the PIN models have indeed learned proper set-input functions, instead of overfitting to the election sizes observed during training. Note that it is impossible to test the MLP models on elections with more than 99 voters due to the fixed input size. Scaling up an MLP model to fit elections with up to 199 voters would require doubling its input size, increasing its number of parameters and retraining. This is a major advantage that PINs have over MLP models. We leave a thorough investigation of PIN models under drastic distribution shifts (such as elections that are orders of magnitude larger) to future work.

**Generalization to real-world datasets** We trained the networks on synthetic data to mimic the classical voting rules, and tested them on three real-world datasets: the Sushi dataset [Kamishima, 2003] (10 candidates), the Mechanical Turk Puzzle dataset [Mao et al., 2013] (4 candidates) and a subset of the Netflix Prize Data [Bennett et al., 2007] (3-4 candidates). We randomly sampled elections from these datasets with number of voters from 2 to 99 for testing. Details of the testing data can be found in Supplementary Material.

The results (Table 4) show that PINs have learned voting rules that generalize well to real datasets, and slightly outperform those learned by the MLP. Interestingly, the mimicking accuracy on the real datasets is higher than that on the synthetic training data (Table 1). This likely indicates that the winners for synthetic elections generated by the uniform Dirichlet distribution are harder for the networks to determine than most real-world scenarios. While we found that it was not necessary in our experiments, it is also feasible to fine-tune synthetically trained networks on real data to further adapt to unseen distributions.

---

[9]PINs may be more efficient in computing the winner of a single election, and more importantly, batches of elections can be parallelized very efficiently in PINs with GPUs. This is a property that traditional solvers lack.

| Arch. | Mimicking Acc. (Netflix) | | | | | Mimicking Acc. (Sushi) | | | | | Mimicking Acc. (MTP) | | | | |
|---|---|---|---|---|---|---|---|---|---|---|---|---|---|---|---|
| | P. | B. | C. | M. | K. | P. | B. | C. | M. | K. | P. | B. | C. | M. | K. |
| Set T. | **1.** | **1.** | **1.** | **1.** | **.99** | **1.** | **1.** | **.98** | **1.** | **.98** | **1.** | **1.** | **1.** | **1.** | **1.** |
| GIN | **1.** | **1.** | .98 | .98 | .97 | **1.** | **1.** | **.98** | .98 | .96 | **1.** | **1.** | .98 | .99 | .99 |
| DeepS. | **1.** | **1.** | **1.** | **1.** | **.99** | **1.** | **1.** | **.98** | .99 | .97 | **1.** | **1.** | **1.** | **1.** | **1.** |
| MLP | **1.** | **1.** | .96 | .97 | .97 | **1.** | **1.** | **.98** | .98 | .96 | **1.** | **1.** | .98 | .98 | .99 |

Table 4: Voting rule mimicking accuracy of networks trained on synthetic elections and tested on three different real-world datasets. Compared to the synthetic data (Table 1), the performance of all learned voting rules are substantially *better* when applied to real data. This compelling zero-shot generalization performance suggests that training on diverse and difficult synthetic datasets is a promising approach to overcome the sample inefficiency of training large neural networks.

Apart from real-world datasets, we show in the Supplementary Material (Table 6 and 7) that the learned voting rules also generalize well to other synthetic data distributions. The strong generalization performance of synthetically trained voting rules (including the MLP baseline) is significant from a sample-complexity perspective. Procaccia et al. [2009] showed that positional scoring rules are efficiently PAC learnable, but learning pairwise comparison based voting rules in general requires an exponential number of samples. While training neural networks requires large amounts of data, our synthetic-to-real generalization results suggest that training on diverse and difficult synthetic datasets can achieve compelling zero-shot generalization performance. This is especially significant for NP-hard voting rules such as Kemeny.

| Architect. | Uniform | | Polarized | | Indecisive | |
|---|---|---|---|---|---|---|
| | Util. | Egal. | Util. | Egal. | Util. | Egal. |
| Set Trans. | 0.68 | **0.66** | 0.67 | **0.68** | **0.70** | 0.54 |
| GIN | **0.69** | 0.65 | **0.68** | 0.67 | **0.70** | 0.55 |
| DeepSets | 0.67 | 0.65 | 0.66 | 0.67 | 0.68 | **0.57** |
| MLP | 0.67 | 0.65 | 0.66 | 0.67 | 0.69 | 0.54 |
| Plurality | 0.42 | 0.40 | 0.47 | 0.46 | 0.39 | 0.31 |
| Borda | 0.56 | 0.58 | 0.50 | 0.51 | 0.61 | 0.53 |
| Copeland | 0.52 | 0.53 | 0.48 | 0.49 | 0.56 | 0.48 |
| Maximin | 0.50 | 0.50 | 0.46 | 0.47 | 0.53 | 0.45 |
| Optimal | 0.65 | - | 0.65 | - | 0.67 | - |

Table 3: The accuracy by which learned voting rules can predict the candidate that maximizes the utilitarian and egalitarian social welfare functions on different utility distributions ("uniform", "polarized" and "indecisive") — higher is better. The PIN models slightly outperform, and mostly are on par with the MLP ones. All the neural network models outperform classical voting rules.

## 4.2 Maximizing social welfare

In almost all real elections, we only have access to voter preference profiles, but not the underlying utilities. While the theoretically optimal voting rule is known assuming utilitarian social welfare (Section 2.3), no theoretical results are yet established for general social welfare functions.

However, with PINs being general function approximators, we can train them to *discover* these potentially unknown social welfare-maximizing voting rules. We define an *oracle* to be the "ideal" voting rule that has access to the *underlying utilities* (possible on synthetic data), and computes the winners that maximize some social welfare function. We then trained the neural networks to mimic the oracles given only the voter preference profiles.

We experimented with two different social welfare functions (utilitarian and egalitarian) and three different utility distributions (uniform, polarized and indecisive). We report the accuracy of the best candidate predictions attained by the learned voting rules in Table 3. We compare these with the performance achieved by classical voting rules (Plurality, Borda, Copeland and Maximin).[10] For the utilitarian social welfare, we report the performance of the theoretically optimal voting rule as well.

Table 3 shows that all the neural network models achieve better accuracy than the classical voting rules. When the theoretical optimal voting rule is known (i.e. with utilitarian social welfare), the

---

[10]Due to the computational cost of generating training data with large numbers of candidates, we do not report Kemeny results here.

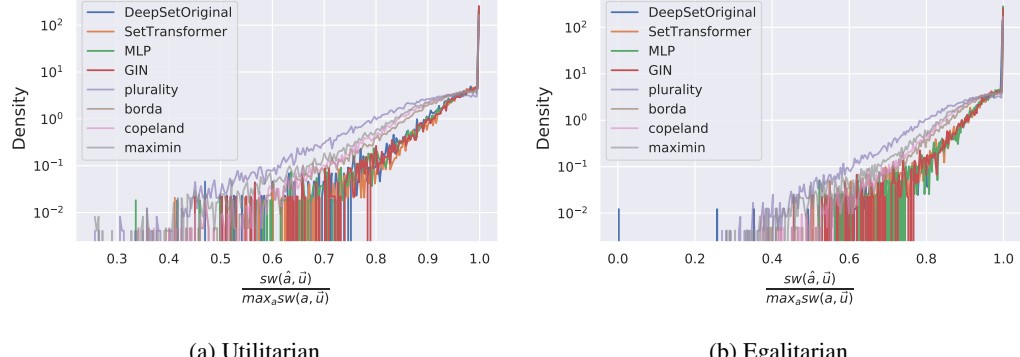

|                     |                     |
| :-----------------: | :-----------------: |
| (a) Utilitarian     | (b) Egalitarian     |

Figure 1: Normalized histogram of the ratio between the social welfare following different voting rules and the optimal social welfare, for the utilitarian and egalitarian social-welfare functions. Data are sampled from the "uniform" distribution.

neural networks match the accuracy of the optimal rule.[11] This indicates that neural networks are highly effective in discovering novel voting rules that maximize general social welfare functions.

Besides the prediction accuracy, we also evaluate the distribution of the resultant social welfare following the baseline and the learned voting rules. Figure 1 shows the normalized histogram of the ratio between the social welfare following a voting rule and the optimal social welfare:

$$\frac{sw(\hat{a}, \vec{u})}{\max_a sw(a, \vec{u})} \quad (\hat{a} \text{ is the winner chosen by a voting rule}).$$

The conclusion from Figure 1 is consistent with the accuracy results. The neural networks models are better at maximizing social welfare than the classical voting rules, in that they achieve higher social welfare values.

Note that although the performance of PINs in Table 3 and Figure 1 is on par with MLP, PINs still have the fundamental advantage of respecting voter anonymity and being able to generalize to arbitrary voter numbers, as discussed in Section 4.1.

# 5 Related works

Procaccia et al. [2009] showed that positional scoring rules are efficiently PAC learnable, but learning pairwise comparison-based voting rules in general requires an exponential number of samples. Boutilier et al. [2015] proved that the optimal voting rule that maximizes the average-case utilitarian social welfare is a positional voting rule for any neutral utility distributions. Also, the line of work in distortion focuses on developing and analyzing voting rules that are optimal for the worst-case utilities [Boutilier et al., 2015, Caragiannis et al., 2017]. We focus on the average-case maximization of social welfare, but without assumptions on the social welfare function or any given utility distribution. Xia [2013] proposed a generic workflow towards designing social choice mechanisms using machine learning and outlined approaches could take to incorporate desirable axioms. Armstrong and Larson [2019] and Firebanks-Quevedo [2020] proposed using deep learning to learn voting rules that satisfy desirable axioms.

Kujawska et al. [2020] and Burka et al. [2021] used several classical machine learning methods such as support vector machines, gradient boosted trees and shallow MLPs to mimic existing voting rules (including Borda, Kemeny and Dodgson). Compared to the models in Kujawska et al. [2020] and Burka et al. [2021], our PIN models have fundamental advantages such as anonymity by construction and generalization to unseen numbers of voters.

---

[11]Note that the theoretically optimal voting rule maximizes expected social welfare, not the accuracy by which the optimal candidate is elected, which explains why its accuracy value may trail that of the learned voting rules.

# 6  Limitations and future directions

Our work is currently limited to elections with complete rankings and strict orders, and to the single-winner case. Future work should explore generalization to rankings that are incomplete and with ties, and potentially to predicting rankings instead of a single winner. Moreover, we're interested in architectures that support an arbitrary number of candidates at test time by construction (for PIN architectures, number of candidates encountered during test time is limited by the number of candidates the network was trained on.) We would also like to explore the reinforcement learning approach to maximizing social welfare rather than supervised learning (discussed in Section 3.3).

Last but not the least, we would like to raise two important points of considerations to the attention of practitioners working on sensitive applications: 1) The neural networks architectures we discuss in the paper are neither interpretable nor transparent by construction, and further effort is needed to understand their inner workings. 2) There are no guarantees as to their worst case performance and further empirical (and, is possible, theoretical) analyses is needed. Until the aforementioned points of consideration are addressed, we caution against the direct application of our methods on safety-critical and sensitive settings.

# 7  Conclusion

We show that PIN architectures are both theoretically and empirically well-suited for learning voting rules. PIN models respect voter anonymity by construction and are universal approximators for representing voting rules. After training on synthetic data, PIN models generalize seamlessly to unseen real datasets and an unseen number of voters. They can also be trained to maximize social welfare better than fixed classical voting rules. The flexibility and effectiveness of our approach clears some of the hurdles in the design and implementation voting rules.

# 8  Acknowledgement

We are indebted to Nisarg Shah for his close guidance and feedback over the course of this work. We would also like to thank Roger Grosse for his helpful feedback on an early draft of this paper.

CA was supported by the CIFAR Graduate Scholarships - Doctorate scholarship. XB was supported by a Natural Sciences and Engineering Research council (NSERC) Discovery Grant. Resources used in preparing this research were provided, in part, by the Province of Ontario, the Government of Canada through CIFAR, and companies sponsoring the Vector Institute.

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
