# A    Background on permutation-invariant neural network architectures

## A.1    Background on graph neural networks

Many GNN architectures iteratively update node features following a neighborhood aggregation scheme. Concretely, a graph network layer has the following structure:

$$a_v^{(k)} = \text{AGG}^{(k)}(\{h_u^{(k-1)} : u \in \mathcal{N}(v)\})$$
$$h_v^{(k)} = \text{COMB}^{(k)}(h_v^{(k-1)}, a_v^{(k)})$$

where $h_v^{(k-1)}$ is the feature vector at node $v$ at layer $k-1$, $\mathcal{N}(v)$ is a function that returns the neighbors of $v$, $\text{AGG}^{(k)}$ is a permutation-invariant function that aggregates the features of the neighbor nodes and $\text{COMB}^{(k)}$ is a function that produces the next layer features by processing the features at the previous layer $h_v^{(k-1)}$ and the result of the aggregation $a_v^{(k)}$ [Xu et al., 2018]. Once the node features are refined through multiple GNN layers, the graph level embedding is obtained by *pooling* all the node features using a permutation-invariant function.

$$h_G = \text{POOL}(\{h_v^{final} : v \in \mathcal{V}\})$$

where $h_v^{final}$ is the final node embeddings and $\mathcal{V}$ is the set of all nodes in the graph.

### A.1.1    Graph Isomorphism Network (GIN)

The GIN architecture updates the node features as follows:

$$h_v^{(k)} = \text{MLP}^{(k)}((1 + \epsilon^{(k)}) \cdot h_v^{(k-1)} + \sum_{u \in \mathcal{N}(v)} h_u^{(k-1)}) \tag{1}$$

Xu et al. [2018] show that the GIN architecture is at least as powerful as the *Weisfeiler-Lehman (WL) graph isomorphism test* [Leman and Weisfeiler, 1968] in distinguishing graph structures. We use this result to prove that the GIN architecture is a universal approximator of set-input functions in Section 3.2.

Instead of just using the last layer node features as inputs to the graph-level pooling operation, GIN uses a concatenation of the node features across *all* layers:

$$h_G = \text{CAT}(\text{POOL}(h_v^{(k)}|v \in \mathcal{V})|k \in \{1 \dots K\})$$

where $K$ is the index of the last GIN layer and CAT is the concatenation operation along the feature dimension.

## A.2    Background on the Set Transformer

The Query-Key-Value (QKV) attention operation takes in three tensors as input (query, key and value tensors of shape $\mathbb{R}^{n_q \times d_{kq}}$, $\mathbb{R}^{n_{kv} \times d_{kq}}$ and $\mathbb{R}^{n_{kq} \times d_v}$ respectively), and produces one tensor as output of shape $\mathbb{R}^{n_q \times d_v}$:

$$\textbf{QKV}(Q, K, V; w) = (\omega(QK^T))V \tag{2}$$

where $\omega$ is a normalization function such as softmax. The the rows of $Q$, $K$ and $V$ can be interpreted as query, key and value vectors, and the $QKV$ operation can be interpreted as implementing a differentiable dictionary retrieval operation. To give the neural network the ability to control both the output space of the QKV operation, as well as the space in which query-key similarities are computed, it is common to pre-and-post-multiply all of the input and output tensors by learnable weight matrices. This gives rise to *multihead attention operation*. This operation is described as follows:

$$\textbf{MULTIHEAD}(Q, K, V; \mathcal{W}, \omega) = \text{CAT}(O_1, \dots, O_h)W^O$$
$$O_j = \textbf{QKV}(QW_j^Q, KW_j^K, VW_j^V; \omega)$$

where $W^Q \in \mathbb{R}^{d_{kq} \times d_{kq}}$, $W_j^K \in \mathbb{R}^{d_{kq} \times d_{kq}}$, $W_j^V \in \mathbb{R}^{d_v \times d_v}$ and $W_j^O \in \mathbb{R}^{d_h * d_v \times d_{out}}$ are the learnable parameters, $\omega$ is the normalization operation, $h$ is the total number of attention heads and CAT is the concatenation operation.

The properties of Multihead attention make it suitable for set-input network architectures:

**Property 3.** *Multihead attention is equivariant under the permutation of the query vectors and invariant under the joint permutation of the key-value vectors [Lee et al., 2019].*

The Set Transformer architecture makes use of these equivariance/invariance properties to build highly expressive permutation equivariant/invariant encoder and decoder networks. Details of these encoder and decoder architectures are described in Section A.2.1.

Set transformers can also *learn* set-level pooling operations using the multihead attention instead of using static operations such as mean or sum-pooling. This can be done by learning *seed vectors* which are used as queries to the Query-Key-Value attention operation [Lee et al., 2019].

### A.2.1 Full Set Transformer architecture

The encoder is comprised of a chain of "Set-Attention-Blocks" (SAB), defined as,

$$SAB(X) := MAB(X, X),$$
$$\text{where } MAB(X, Y) = LayerNorm(H + rFF(H)),$$
$$H = LayerNorm(X + Multihead(X, Y, Y)).$$

Here, $LayerNorm$ stands for the Layer Normalization operation proposed by [Ba et al., 2016] and $rFF$ stands for a single layer of feed-forward neural network simultaneously applied to the value vectors.

The decoder is comprised of a chain of Self-Attention Blocks, as well as "Pooling by Multihead Attention (PMA)" operation that are defined as $PMA_k(Z) = MAB(S, rFF(Z))$ where $S \in \mathbb{R}^{kxd}$ is a learned query matrix.

The full encoder and decoder architectures can be characterized as follows:

$$\text{SetTrans}(X) = \underbrace{rFF(SAB(PMA_k}_{decoder}(\underbrace{SAB(SAB(X))}_{encoder}))) \tag{3}$$

In our experiments, we used 4 layers of $SAB$ blocks inside the encoder instead of 2.

## B Proof of Theorem 1 - universality of GIN

*Proof.* To prove the theorem, it is sufficient to show that for any fully-connected, non-isomorphic graphs $G_1$ and $G_2$, the Weisfeiler-Lehman test will decide them as non-isomorphic. In other words, even though the WL test is not complete for the general graph isomorphism problem, it is complete when the graphs are fully-connected.

Let $v_i$, $v_j$ be two nodes in a fully-connected graph $G$. Because they share all other neighbours in $G$, $v_i$ and $v_j$ will have the same label in the first iteration of the WL test if and only if their initial labels (i.e. features) are equal. This implies that the WL test for fully-connected graphs terminates after the first iteration. The WL test decides $G_1$ and $G_2$ are non-isomorphic if and only if the multi-sets of their node features are different, which for fully-connected graphs is equivalent to $G_1$ and $G_2$ being non-isomorphic. $\square$

## C Input-output representations

To convert preference profiles $\vec{\sigma} = (\sigma_i, \ldots, \sigma_n) \in (S_m)^n$ into a representation that can be fed in set-input neural networks, we first define a *candidate id function* $c : A \mapsto \mathbb{R}^{n_c}$ that maps each alternative to a unique $n_c$ dimensional real-valued vector. An obvious choice of $c$ is assigning a unique integer between 0 and the number of candidates to each alternative, in which case $n_c$ is equal to 1. Another alternative is assigning a unique one-hot vector to each candidate, in which case $n_c$ is at least as large as the number of candidates. Using the candidate id function, we then compute *vote vectors* $v_i = [c(\sigma^{-1}(a_1)), \ldots, c(\sigma^{-1}(a_m))] \in \mathbb{R}^{(m \cdot n_c)}$ for each candidate and finally concatenate the vote vectors to prepare the input that can be fed to a neural network: $X = [v_1, \ldots, v_n] \in \mathbb{R}^{(n \times m \cdot n_c)}$.

PINs support elections with an arbitrary number of voters; however, the vote vectors they receive must have a fixed dimensionality. Therefore, if the sampled election has lesser number of candidates

than the maximum number the network is trained to support, we simply zero-pad the *vote vectors*, so that they have a fixed dimensionality. When training fixed-input architectures such as multilayer-perceptron for benchmarking, we also use zero-padding along the voter dimension.

While information theoretically equivalent, the one-hot candidate ids approach often outperforms integer ids in our experiments. This is likely due to the fact that it avoids imposing an arbitrary ordering between the candidates.

The output of the neural networks $\hat{y} \in \mathbb{R}^m$ can be used to compute each candidate's probability of winning. For representing single-winner elections, the probability of candidate $a_j$ being the winner can be computed using $Pr[\text{winner} = a_j] = [\text{softmax}(\hat{y})]_j$. For multiple winner elections, the probability of candidate $a_j$ being one of the winners can be computed using $Pr[a_j \in \text{winners}] = \text{sigmoid}([\hat{y}]_j)$. In this paper, we only work with single-winner elections.

### C.1 Anonymity and neutrality of learned rules

If one uses the input representation described above, PINs will automatically learn anonymous voting rules. While this might be a desirable property for many use cases, certain applications — such as the ones where voters have persistent identities as in recommender systems — might require violating this property. This can easily be achieved by appending the vote vectors with unique "voter id" vectors so that the neural network can process which ranking was submitted by which voter.

Neutrality (the property of being agnostic to candidate identities) is also easy to achieve using PINs by simply re-tiling the input tensor so that the network is permutation-invariant across the candidate dimension. However, this strategy prevents the network from being anonymous. We focus on anonymous architectures in this paper.

## D  Experiment details

### D.1  Network architecture details

We now explain the network architectures we used in our experiments in depth.

**DeepSets:**   Both the encoder and decoder networks have a 5 layer fully connected structure with the LeakyReLU activation [Maas et al., 2013] and LayerNorm [Ba et al., 2016] normalization. The width of the hidden layers were set at 1065 neurons so that the whole network has roughly 10 million parameters in total.

**GIN:**   We used the GIN implementation provided by the Deep Graph Library [Wang et al., 2019] without much modification in our experiments. The whole network had 6 neighborhood aggregation layers. Each aggregation layer used a different 2 layer fully connected network with ReLU activations and LayerNorm normalizer [Ba et al., 2016].[12]  We also set the $\epsilon$ parameter (the weighing of the node's own features before the aggregation scheme) to be learnable. We used the sum pooling inside the aggregation scheme, and the mean pooling scheme for the graph-level readout. We did not use Dropout [Srivastava et al., 2014] as originally proposed by the authors. We set the hidden layer with to be 995, so that the whole network had roughly 10 million parameters.

**Set Transformer:**   We used an encoder with 4 self attention blocks, and a decoder with 1 pooling by multihead attention (PMA) layer followed by another self-attention layer. Our self-attention block implementation differs slightly from the one the authors provide [Lee et al., 2019] in that it makes use of the "pre-layer norm" self-attention block configuration [Xiong et al., 2020], which helped improve training stability. We used LayerNorm as the default normalized and the ReLU activation throughout the network. We used 20 heads in all attention operations and computed query-key similarities in a 28 dimensional space. When added together, this results in roughly 10 million parameters.

---

[12]We replaced all the Batch normalization layers [Ioffe and Szegedy, 2015] in the original GIN architecture with Layer normalization. We empirically confirmed that this modification did not lead to any performance degradation, and led to improved training stability.

**MLP:** We used a standard 5 layer fully connected network with ReLU activations and LayerNorm normalizer. Due to the very large input dimensionality (83259 units), most of the parameters of this network are stored in the first layer. Setting the hidden layer width to be 120 results in the model having roughly 10 million parameters.

**Training Setup** We ran experiments with the Lookahead optimizer [Zhang et al., 2019] with its default hyperparameters, and found that it significantly helps with training DeepSet models, while its effect on other architectures is minimal. We didn't use any form of regularization — at no point in our experiments we observed overfitting behaviour thanks to the infinite supply of synthetic data. We also clipped gradients whose L2 norm surpassed 1 for increase training stability. We trained all of the networks using the PyTorch framework [Paszke et al., 2017], on NVIDIA T4 GPUs. Depending on the task and model, each training run took about 1 to 8 days to complete.

## D.2 Real-world test data details

**Netflix Prize Data datasets [Bennett et al., 2007]** Netflix Prize Data were collected by encouraging Netflix subscribers to rate the movies they watch, and express how much they liked or disliked them. We used the first ten elections in the Netflix Prize Data dataset.[13] These elections have 3 candidates, and 448 - 1860 voters. The dataset contains the complete ranking information in strict order for each election. Since we aim to evaluate the generalization performance on different data distributions, rather than on unseen number of voters, we sub-sampled each of the elections to contain 2-99 voters (the same numbers of voters that the networks are trained on). For each of the ten elections, we generated 16,384 "sub-elections" of voter size 2 to 99 via random sampling. In total, the test dataset contains 163,840 elections.

**Sushi dataset [Kamishima, 2003]** The sushi dataset was collected by surveying 5000 individuals for their preferences about various kinds of sushi.[14] We used the "Sushi 10 rank" dataset, which contains the 5000 individuals' complete strict rank orders of 10 different kinds of sushi. We randomly sampled 16,384 "sub-elections" of voter size 2 to 99 for our experiments.

**Mechanical Turk Puzzle datasets [Mao et al., 2013]** The datasets were collected using Mechanical Turk, and contain 4 elections, each with 4 candidates and 793-797 voters.[15] For each of the 4 elections, we randomly sampled 16,384 "sub-elections" of voter size 2 to 99. In total, the test dataset contains 65,536 elections.

# E Additional experiment results

## E.1 Mimicking existing voting rules

### E.1.1 Similarity of traditional voting rules

Table 5 shows the accuracy of predicting the winner of a traditional voting rule using another traditional voting rule.

### E.1.2 Zero-shot generalization to other utility distributions

Table 6 and 7 show the zero-shot generalization performance of learned voting rules that are trained on the uniform utility distribution, but tested on other utility distributions (polarized, indecisive).

### E.1.3 Generalization to Unseen Numbers of Voters

We include additional experiment results for generalization to unseen numbers of voters in this section. The performance of the PIN models on mimicking the Borda voting rule is included in the

---

[13]We downloaded the election data from `https://www.preflib.org/data/election/netflix`. Licensing information of the Netflix Prize Data can be found at `https://www.kaggle.com/netflix-inc/netflix-prize-data`.

[14]We downloaded the data from `https://www.preflib.org/data/election/sushi`. Licensing information can be found at `https://www.kamishima.net/sushi`.

[15]We downloaded the datasets from `https://www.preflib.org/data/election/puzzle`.

| Target | Alternative | | | |
|---|---|---|---|---|
| | Plurality | Borda | Copeland | Maximin |
| Plurality | 1.0 | 0.387 | 0.383 | 0.385 |
| Borda | 0.298 | 1.0 | 0.736 | 0.654 |
| Copeland | 0.355 | 0.815 | 1.0 | 0.771 |
| Maximin | 0.358 | 0.769 | 0.817 | 1.0 |
| Kemeny | 0.511 | 0.754 | 0.828 | 0.911 |

Table 5: The accuracy of predicting the winner of a traditional voting rule ("target") using another traditional voting rule ("alternative"). Each value is computed using 16,384 sampled elections from the synthetic training distribution. Note that 1) Kemeny is not used as an alternative voting rule, because its computation complexity prevents it from being applied to the election sizes of other target voting rules (up to 29 candidates); 2) the fact that the table is not symmetric is an artifact of tie elimination in our data generation process. When a tie is present using the target voting rule, we discard the election and repeat the sampling process until a tie is not present. This causes a practical disparity among the data distributions for different target voting rules and results in the asymmetry of this table.

| Architect. | Mimicking Accuracy | | | | |
|---|---|---|---|---|---|
| | Plurality | Borda | Copeland | Maximin | Kemeny |
| Set Trans. | 1.000 | 0.985 | 0.828 | 0.794 | 0.928 |
| GIN | 1.000 | 0.989 | 0.811 | 0.772 | 0.816 |
| DeepSets | 1.000 | 0.960 | 0.811 | 0.761 | 0.900 |
| MLP | 0.995 | 0.945 | 0.810 | 0.759 | 0.763 |

Table 6: Mimicking accuracy of learned voting rules trained on the uniform utility distribution, but tested on the polarized utility distribution.

main paper (Table 2). We include the generalization results for mimicking the Plurality, Copeland and Maximin rules in Table 8, 9 and 10.

## E.2 Maximizing social welfare

We show the histograms of the ratio between the social welfare following a voting rule and the optimal social welfare for the "polarized" and "indecisive" distributions in Figure 2 and 3.

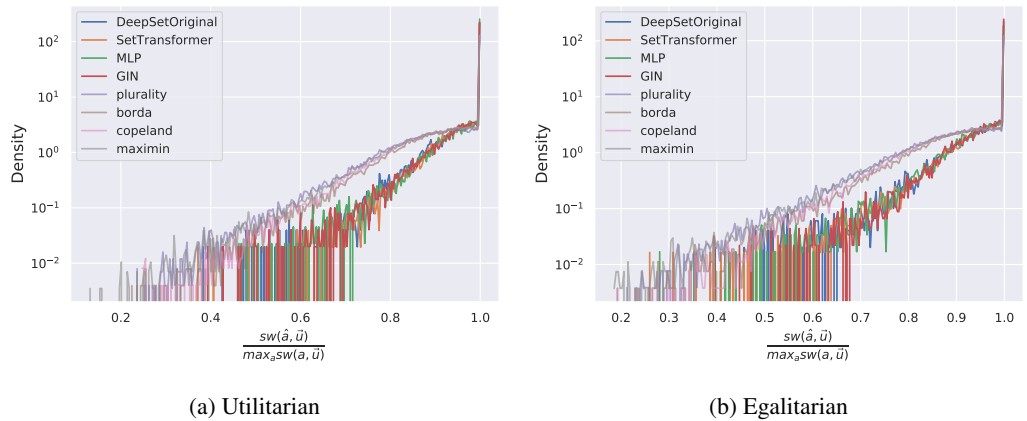

(a) Utilitarian

(b) Egalitarian

Figure 2: Normalized histogram of the ratio between the social welfare following different voting rules and the optimal social welfare, for the utilitarian and egalitarian social-welfare functions. Data are sampled from the "polarized" distribution.

| Architect. | Mimicking Accuracy | | | | |
|---|---|---|---|---|---|
| | Plurality | Borda | Copeland | Maximin | Kemeny |
| Set Trans. | 1.000 | 0.984 | 0.813 | 0.797 | 0.926 |
| GIN | 1.000 | 0.990 | 0.807 | 0.775 | 0.814 |
| DeepSets | 1.000 | 0.963 | 0.809 | 0.784 | 0.889 |
| MLP | 0.996 | 0.943 | 0.796 | 0.762 | 0.759 |

Table 7: Mimicking accuracy of learned voting rules trained on the uniform utility distribution, but tested on the indecisive utility distribution.

| Architect. | Mimicking Accuracy for Different Number of Voters | | | |
|---|---|---|---|---|
| | (within-domain) | | (out-of-domain) | |
| | 2-49 | 50-99 | 100-149 | 150-199 |
| Set Trans. | 1.0 | 1.0 | 1.0 | 1.0 |
| GIN | 1.0 | 1.0 | 1.0 | 1.0 |
| DeepSets | 1.0 | 1.0 | 1.0 | 1.0 |
| MLP | 1.0 | 0.99 | N/A | N/A |

Table 8: The test accuracy of permutation-invariant networks on guessing the Plurality winner with different number of voters. The networks are trained with 2-99 voters.

| Architect. | Mimicking Accuracy for Different Number of Voters | | | |
|---|---|---|---|---|
| | (within-domain) | | (out-of-domain) | |
| | 2-49 | 50-99 | 100-149 | 150-199 |
| Set Trans. | 0.83 | 0.81 | 0.81 | 0.80 |
| GIN | 0.82 | 0.80 | 0.80 | 0.80 |
| DeepSets | 0.84 | 0.82 | 0.80 | 0.79 |
| MLP | 0.81 | 0.79 | N/A | N/A |

Table 9: The test accuracy of permutation-invariant networks on guessing the Copeland winner with different number of voters. The networks are trained with 2-99 voters.

| Architect. | Mimicking Accuracy for Different Number of Voters | | | |
|---|---|---|---|---|
| | (within-domain) | | (out-of-domain) | |
| | 2-49 | 50-99 | 100-149 | 150-199 |
| Set Trans. | 0.83 | 0.77 | 0.76 | 0.74 |
| GIN | 0.79 | 0.74 | 0.73 | 0.72 |
| DeepSets | 0.81 | 0.75 | 0.73 | 0.74 |
| MLP | 0.79 | 0.73 | N/A | N/A |

Table 10: The test accuracy of permutation-invariant networks on guessing the Maximin winner with different number of voters. The networks are trained with 2-99 voters.

| Architect. | Mimicking Accuracy for Different Number of Voters | | | |
|---|---|---|---|---|
| | (within-domain) | | (out-of-domain) | |
| | 2-49 | 50-99 | 100-149 | 150-199 |
| Set Trans. | 0.94 | 0.94 | 0.93 | 0.92 |
| GIN | 0.82 | 0.81 | 0.79 | 0.76 |
| DeepSets | 0.91 | 0.89 | 0.89 | 0.87 |
| MLP | 0.78 | 0.77 | N/A | N/A |

Table 11: The test accuracy of permutation-invariant networks on guessing the Kemeny winner with different number of voters. The networks are trained with 2-99 voters.

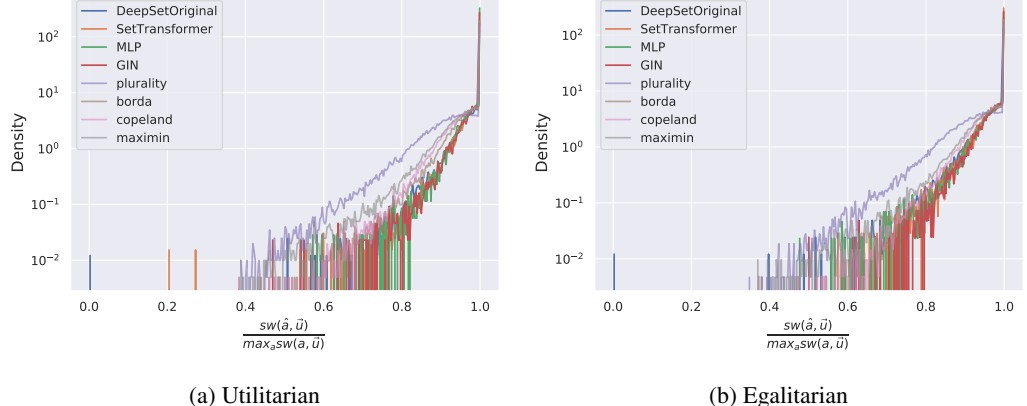

(a) Utilitarian

(b) Egalitarian

Figure 3: Normalized histogram of the ratio between the social welfare following different voting rules and the optimal social welfare, for the utilitarian and egalitarian social-welfare function. Data are sampled from the "indecisive" distribution.