# OpenReview forum: "Learning to Elect"
_NeurIPS.cc/2021/Conference — NeurIPS 2021 Poster_

### Official Review · Reviewer_sGWw · 2021-07-10

**Rating:** 7
**Confidence:** 4

**Summary:**

The Authors report a set of experiments with using neural networks with a particular architecture to learn optimal voting rules. They provide a concise theoretical description of the problem and prove that the chosen type of NNs are universal approximators of voting rules. They demonstrate experimentally that NNs can be trained using voter's ranked preferences only.

The experiments are of two kinds:

1. teaching the NN known classical voting rules

2. learning a new rule to maximise a given social welfare function

Their results show that NNs can learn known voting rules very well and that they can discover new voting rules which select optimal candidates more often than classical voting rules.

--- EDIT ---

Increased the score to 7 after the clarifications.

**Limitations And Societal Impact:**

The Authors comment on the societal impact, but the topic needs to be explored further, especially if the learned rules were to be used in a public setting (e.g. political elections).

For limitations, see pt. 3 above.

**Main Review:**

The Authors tackle an interesting problem and report encouraging results. The paper is clearly written and the theoretical results are explained well. I liked the fact that the Authors made the effort to provide an introduction to the problem domain for lay readers (like me). However, there are some problems which prevented me from giving a higher score:

1. the notation is not always clear:

- is vec{u} a vector of one voters' utilities per candidate, or a vector of utility functions of all voters? Algorithm 1 suggests the 2nd, but the caption under Figure 1 suggests the 1st. This reviewer is a bit confused.

- what exactly is the prediction loss function l optimised by Algorithm 1?

2. the real-world datasets are not described sufficiently even in the Supplementary Material. Since the synthetic datasets used in the paper contain only uniform distributions of preferences (every candidate is equally likely to win on average), it would be interesting to know if the real-world datasets deviate from that. For example, is there a sushi type which is much more popular than all others? Many real-world situations such as presidential elections contain two or three dominant politicians and a string of niche candidates with little hope of winning.

3. the discussion of the limitations of the method is lacking:

- firstly, usually we deal with more than 99 voters. The Authors do not comment at all on the question how well would their method scale to realistic numbers of voters (i.e. up to millions).

- secondly, since we do not know the actual utility functions of the voters, we probably need voting rules which are robust to perturbations. The Authors did not seem to have tested whether a voting rule learned on one type of utility functions behaves "decently" on other utility functions (e.g. train on "uniform", test on "polarized"). In other words, the network may be overfitting to modelling assumptions.

- thirdly, the Authors do not comment on the question of interpretability of learned voting rules - an important problem if we are to implement such a rule in public life. It would also be fascinating to understand how the learned rules work and how they differ from the classical ones. E.g. how often does a learned rule indicate a different winner than the classical ones?

- finally: as far as I understand, the presented NN architectures do not fulfil the neutrality criterion (as discussed in Appendix C.1 - I think should be moved to the main text). How do we know that, for datasets which have a dominant candidate, the network doesn't simply memorise which candidate should win? Couldn't this be tested by randomly permuting the candidates in the test set? (maybe the Authors have done that? i'm not sure)

4. Figure 1 is hard to understand for me. It seems that the histograms are heavily concentrated in 1, with only the tails differing. Why not also provide the maximised value and compare it with the theoretical maximum? It would be much easier to understand than the histogram. (Especially that the meaning of \vec{u} is not clear.) The same comment applies to similar Figures in Suppl. Mat.

5. Minor: At least a sketch of the proof of Theorem 1 should be present in the main text.

Some of the above comments are probably impossible to address given the page limits, so the Authors should treat them as my wishlist for their further work.

**Time Spent Reviewing:**

5

---

> ### Author Response · Authors · 2021-08-10
> **Thank you for your review**
>
> Thank you for your comments and your questions! We appreciate that you found the problem we’re tackling interesting and our results encouraging. We’ll try to address the questions you listed:
> * __Notation:__
>     * __Meaning of \vec{u}:__ $\vec{u} \in \mathbb{R}^{n\times m}$ is the utility profile, or the utility functions of all voters (line 76). In Figure 1, the social welfare function takes the utility functions of all voters as an argument. We’ll improve the writing to avoid confusion.
>     * __Loss function l:__ In terms of optimization, any convex loss function that has its minima at (\hat{y} = y) is suitable here. We use the cross entropy loss in our experiments (i.e. set the label to be y). We will clarify this in the revised version of the paper.
> * __Description of real-world datasets:__
> Thank you for the suggestion to further describe the real-world datasets. It’s indeed the case that in real-world situations, the utility profile distribution is not symmetric to each candidate. In the sushi dataset for example, the most preferred sushi types include uni (sea urchin) and toro (fatty tuna). Although since we sample many subsets from these real-world datasets for testing, the exact winner might not be consistent and will depend on the voting rule. In general, our synthetic datasets are often more challenging for learning voting rules than the real datasets. A network that performs well on the synthetic datasets, our experiments show, can be expected to do well on real ones too.
> * __Discussion of limitations:__
>     * __Large number of voters:__ While, in principle, our techniques can be applied on much larger elections, how to process and generalize to _very_ large set sizes is an open question - not just in the area of learning voting rules, but also generally in sequence modelling and natural language processing. We’ll elaborate on this in our revision.
>     * __Different utility distributions:__  As mentioned in our general response, we tested how well PIN architectures trained on elections sampled from uniform voter utility distributions zero-shot generalize to other synthetic distributions, such as polarized and indecisive. The results remained consistent across distributions (i.e. ± 1 percentage point differences). This indicates that the networks do exhibit some form of “algorithmic generalization” instead of overfitting to particular distributions.
>     * __Interpretability:__ We’ll make it clear in our revision that we’re not learning voting rules that are also interpretable by construction. We find this to be an exciting future direction - both learning rules that are interpretable by design, or find interpretable approximations to the rules we learn. It is also worth mentioning that there are many other applications of voting rules (like recommender systems) where a black-box system with superior performance (in metrics such as social welfare) could be preferable to an interpretable yet weak system.
>     * __Neutrality:__ Although our NN architectures are not neutral by construction, the distributions we use to sample the training data are symmetric to candidates (i.e. for any sampled election, a different election with permuted candidate ids are equally probable, as explained in section 4). Moreover, the possibility of memorization is practically nonexistent, since we always generate our training data online by sampling from the distributions described in the paper and the likelihood of encountering the same training input multiple times is very small.
> * __Points on clarity:__
>     * __Figures:__ Thank you for pointing this out. We’ll try to improve the clarity of the figures. By plotting the histograms, we wanted to observe how close the networks (or classical voting rules) get to optimizing social welfare when they output sub-optimal candidates. The histograms indicate that the learned voting rules frequently do a better job optimizing social welfare, even when they don’t output the optimal candidate.
>     * __Sketch of proof:__ We’ll try to incorporate a brief description of our proof strategy in the paper.

---

> > ### Comment · Reviewer_sGWw · 2021-08-11
> > **Thank you for the clarifications and additional experiments**
> >
> > I want to thank the Authors for the explanations. I think what they wrote in response to my question about the histograms should be included in the Figure's descriptions.
> >
> > I must have missed the point about permuting candidates while sampling, which will prevent memorising the winner. Perhaps it's worth spelling it out for the reader.
> >
> > A side comment: I am not sure if I agree with the Authors' opinion that in recommender systems interpretability is not important. It probably is less exposed than for elections, but for the providers of recommender systems it sure matters whether they can quickly understand why their system malfunctions or causes harm (e.g. recommending fake news or spreading disinformation about COVID vaccines). It will also matter for the industry regulators (if such exist). One of the reasons why Deep Learning is often treated with distrust in industries like finance is the lack of interpretability, which makes debugging systems difficult.
> >
> > After the clarifications, I'm happy to increase my score to 7.

---

### Official Review · Reviewer_Sk4S · 2021-07-13

**Rating:** 7
**Confidence:** 3

**Summary:**

This paper empirically studies the ability of PIN (Set Transformer, GIN, DeepSets and MLP) applied to predicting the (top-1) winner of voting rules (Plurality, Borda, Copeland, Maximin and Kemeny) and predicting the social-welfare-maximizing (utilitarian and egalitarian) candidates. The PINs are trained on synthetic data, agents whose utility over candidates are generated from Dirichlet distributions. The PINs are tested on synthetic data (Table 1 & 3), synthetic data with more voters than in training time (Table 2) and real world data (Table 4).

**Limitations And Societal Impact:**

The authors have adequately addressed the limitations.

**Main Review:**

Pros:
 - The papers is well written and well motivated. The question of "why would anyone want to predict the outcome of voting rule" is addressed out-front on line 38, and the question of "why not compare the full rankings" is acknowledged in Section 6.
 - The result is strong and surprising (that training on synthetic data generalizes to real-world data).
 - The code is well organized and easy to follow, although I did not attempt to replicate the results.

Cons / questions:
 - Since the voting rules are traditionally assumed to output a ranking (possibly admitted by some scores), I wonder how the PINs do when evaluated in terms of rankings. Strictly speaking, one can use the binary-predicting PINs for n-1 rounds to predict a full ranking, no?
 - On the surprising result that training on synthetic data generalizes to real-world data: does this say anything about the real-world data, that it may follow some Dirichlet distributions?

===============

I acknowledge and appreciate the author response.

**Time Spent Reviewing:**

5 hrs

---

> ### Author Response · Authors · 2021-08-10
> **Thank you for your review**
>
> Thank you for your encouraging review! We appreciate that you found our paper well-written and our results strong and surprising. We’ll try to address the cons / questions:
> * __Output rankings:__ We believe that modifying our method to output rankings is a very promising future direction. Our learning framework is very suitable to do so - all one needs to do is find a differentiable divergence function that computes how “far away” two rankings are, and the rest of the framework can remain the way it is. There already exist methods to do this [1], and we’re looking forward to seeing how well one can learn to output rankings. (Do you mind clarifying what you meant by “using binary-predicting PINs for n-1 rounds”? We wanted avoid speculating in our review)
> * __Connection between real and synthetic datasets:__ This is a very interesting question. While we don’t have definitive answers, we speculate that the reason we can generalize from synthetic datasets to real world ones is that the synthetic dataset is actually much more challenging (unlike what happens with other domains, such as computer vision). Sampling candidate utilities uniformly means that most of the sampled elections will be very close-calls. A network that does well on very close elections can be expected to generalize to real world elections where the winner is much more obvious.
>
>
> [1] Cuturi, Marco, Olivier Teboul, and Jean-Philippe Vert. "Differentiable ranks and sorting using optimal transport." arXiv preprint arXiv:1905.11885 (2019).

---

### Official Review · Reviewer_Csu6 · 2021-07-16

**Rating:** 6
**Confidence:** 3

**Summary:**

The authors study how to learn voting rules, both in terms of mimicking existing voting rules and learning new voting rules to (in particular) maximize different versions of social welfare (e.g., egalitarian or utilitarian social welfare). They show that a particular class of neural network architectures (PINs) are well-suited to this task and evaluate their proposed approach experimentally.


**Limitations And Societal Impact:**

Yes

**Main Review:**

The authors mainly investigate how to mimic different voting rules. Although most of them are relatively straightforward to compute, they do achieve good results mimicking the Kemeny rule, which is a good sign for PINs.

Their theoretical result also remains a bit weak. If I understand it correctly, it states that different permutations are mapped to different embeddings, and doesn't say anything about the feasibility of learning these embeddings.

Overall, I appreciate the fact that the authors used more datasets and included results about the Kemeny rule, which is considerably more complex than other rules they mimic. I still do not know if I find their results incredibly compelling -- it seems like the bulk of the paper is about mimicking voting rules, and only section 4.2 talks about learning rules in order to maximize social welfare, which seems like a much more interesting objective overall.

**Time Spent Reviewing:**

2

---

> ### Author Response · Authors · 2021-08-10
> **Thank you for your review**
>
> Thank you for your helpful feedback!
>
> * __Theoretical results:__ Intuitively, our theorem 1 says that GNNs are able to (have the representational power to) map different inputs (voting profiles) to different embeddings. We then invoke Theorem 2 in [Chen et al, 2019], which states that with two additional feed-forward layers, a graph isomorphism-discriminating GNN becomes universal approximating. Indeed, the universality results do not comment on practical performance, which is why we show extensive empirical results.
>
> * __Value of mimicking existing voting rules:__ Maximizing the social-welfare is definitely an interesting direction (both for this paper and future work). The reason that we devote a significant portion of the paper to mimicking existing voting rules is that it is a necessary condition for learning-based approaches to have general applicability in the area of representing voting rules. We believe that it is not self-evident that stochastic gradient descent should be able to train networks to represent highly discrete decision rules, and our positive results send a positive signal to the social choice community that this research direction has promise.

---

### Official Review · Reviewer_EcCg · 2021-07-18

**Rating:** 5
**Confidence:** 4

**Summary:**

The authors study the problem of using machine learning techniques to 'automatically' learn voting mechanisms. Specifically, the authors use three permutation-invariant neural network architectures: DeepSet, Set Transformer and Graph Isomorphism Network and show that they can learn voting rules as well as maximize certain social welfare functions.

The first contribution is to show that these networks are universal function approximators of permutation invariant functions. Voting protocols typically fall within this class, as a permutation of votes should not lead to different outcomes: in other words, the identities of the voters do not matter.

The second contribution is to develop a learning procedure and evaluate the learned voting rules produced by these three algorithms (as well as a standard multi-layer perceptron, for reference). The way the training worked is to consider a population of voters and candidates such that the utility each voter has for the candidates is sampled from a distribution. Specifically, if there are $n$ voters and $k$ candidates, the utility of each candidate for voter $i$ is sampled by a Dirichlet distributions with all $k$ parameters set to $\alpha_0$. Different values of $\alpha_0$ correspond to different voters: some may prefer a single candidate or many of them equally. These values are then converted into an ordering, which the voters submit to the learned mechanism. The mechanisms' output is compared with the voting rule we are trying to learn (or the option which maximizes social welfare) a loss is incurred and a gradient descent step follows. On synthetic data, the learned mechanisms perform very well for all voting protocols tested, which are: Plurality, Borda, Copeland, Maximin and Kemeny.

Finally, the same network architectures perform even better on real data sets, such as the Netflix Prize Data.

**Limitations And Societal Impact:**

The authors adequately explained the limitations and societal impact.

**Main Review:**

The article is very well written, with a clear exposition of the techniques and related work. It is easy to follow and honest: the results claimed in the abstract and introduction are indeed the ones developed later and their analysis is thorough (except possibly for one point).

The main criticism I have (and the reason I cannot fully recommend the paper) is that I am not totally sure of the following two: 1) why would it be difficult to learn voting rules (or maximize welfare) using off-the-shelf ML techniques? 2) Why would anyone try to re-learn any existing voting rule?

I understand that the answer to the second is that perhaps the network will be able to learn some new voting rules which are not described yet. But how would this scenario appear? We would need some designer of a voting protocol with the capacity to generate hundreds of thousands of votes and outcomes, but without the capacity to formally define their relation. Moreover, the learned voting protocol would be quite difficult to motivate in a real election, since it's internals might be hidden behind the network weights.

Finally, why are the experiments not run for more than 199 voters? I expected the real data sets to run in the hundreds of thousands, but I may have missed something.



**Time Spent Reviewing:**

4

---

> ### Author Response · Authors · 2021-08-10
> **Thank you for your review**
>
> Thank you for your helpful review! The criticisms raise very good questions, which we address below:
>
> * __Why would it be difficult to learn voting rules using off-the-shelf ML techniques?__
> There are two main reasons that it’s difficult for standard off-the-shelf ML techniques to learn voting rules:
>     * The space of voting rules that might have practical relevance (both existing and yet to be discovered ones) involve rules with very complex input-output relationships. For example, the Kemeny voting rule is NP-hard for elections with more than 4 candidates. Most of the off-the-shelf ML techniques we’re aware of don’t have the representational capacity and corresponding learning algorithms to handle this difficult space of functions.
>     * Also importantly, we would like the learned voting rules to accept a variable number of voters. Standard neural networks (like MLPs) don’t satisfy this criterion. We overcome this difficulty using permutation-invariant neural networks (which also have high expressive power to express any set-input function).
>
> * __Why would anyone try to re-learn existing voting rules?__
> Being able to learn existing voting rules in practice is a necessary (but not sufficient) condition for learning-based approaches to have general applicability in the area of representing voting rules. Learning existing voting rules also yields useful insights. For example, our experiments into approximating the Kemeny rule helps us establish that set transformers (among other PIN architectures) have the expressive power to represent very complex voting rules. We also believe that it is not self-evident that stochastic gradient descent should be able to train networks to represent highly discrete decision rules. The promising results shown in this paper could be meaningful to the NeurIPS community.
>
> * __Usefulness of “black box” novel voting rules represented by neural networks__
> Discovering voting rules that are at the same time interpretable is, we believe, beyond the scope of our current investigation. However, we find this to be an exciting future direction, and one that the NeurIPS community could be interested in building on. We also would like to stress that voting systems have applications beyond political elections, or elections within small groups of people. They also have applications in recommender systems, product design and other areas where a black-box system with superior performance (in metrics such as social welfare) could be preferable to an interpretable yet weak system.
>
> * __Zero-shot generalization to a larger number of voters__
> Our main purpose of running these generalization experiments was to test whether the PIN networks were displaying some form of _algorithmic generalization_, as opposed to overfitting to different voter numbers. Training on smaller than 100 voters and achieving roughly the same performance on 200 voters suggests that it is indeed the algorithm that the networks are approximating (please also see our general response).  We leave generalizing to very large elections as future work.

---

### Official Review · Reviewer_9xsn · 2021-07-22

**Rating:** 7
**Confidence:** 4

**Summary:**

The paper studies two problems related to using neural networks as single winner voting rules. (1) Training networks to mimic well-known existing rules, and (2) training networks to act as novel rules that select utility-maximizing alternatives.

The major novel contribution of the paper is in their choice of network architecture. "Permutation-invariant networks" are used, which are agnostic to the order of input data, thus ensuring that the resultant voting rules are anonymous by default.

Experiments show that the 3 PIN architectures considered are able to learn the studied voting rules with high accuracy after a large amount of training. These results are shown for testing done on each of real and synthetic data. Similarly, when the networks are trained with an oracle to learn the welfare maximizing alternative they perform quite well.

**Ethical Concerns:**

No concerns

**Limitations And Societal Impact:**

Yes, the authors very briefly point out the major limitation of this work: there are no clear worst-case performance guarantees so relying on this in practice may be dangerous.

**Main Review:**

Originality:
Only a handful of papers have studied the problem of using neural networks as voting rules and have typically focused on relatively basic MLP models. The novel contribution this paper makes is to demonstrate the importance of considering more complex models. This is also (to my knowledge) the first paper to learn voting rules to optimize social welfare.

Quality:
The minor theoretical contribution of the paper seems sound.
The experimental results make a compelling case for the benefit of their contributions. Some experiments could be more thorough or would benefit from more discussion.

Clarity:
The paper has only a few minor typos and is clearly written. Most relevant details are included, though I would be surprised if they were enough to replicate the paper (although the included source code would presumably help with that and has a readme file that appears to have good instructions for running the code)

Significance:
The focus on more complex network architecture is very likely to be built upon by other researchers and appears to be an advance in the state of the art. Demonstrating the learnability of social welfare maximizing candidates, while not a particularly surprising result to me, is novel and likely to be replicated by other researchers in the future.

Minor notes for clarification or consideration:
- explanation of Simpson's rule in 2.2 could be phrased much more clearly
- 2.3 seems to have limited relevance to the remainder of the paper. Consider making the link more clear or cutting it to expand other sections
- My preference would be to see the input-output representation very briefly discussed in the text (footnote 4), it has been an important focus of other papers in this area and would be of interest to some readers
- What is $\alpha$ in section 4? Is there a reason for not giving the exact value? Is it chosen randomly? (If so, how?)
- I can't find details on the distribution of numbers of voters and candidates within a training set. I believe a single set contains examples with varying numbers of voters (distributed how?), I'm not sure if the number of candidates varies. If so, that would be worth emphasizing as other work relies on a fixed number of candidates.
- Is there a reason for training with 2 candidates? All rules will essentially reduce to plurality at that point.
- Does training with varying numbers of voters help generalization to larger voter populations? ie. Would Table 2 have the same out-of-domain results if the networks were trained on elections with a constant number of voters? The answer might give insight as to whether the training procedure or architecture is responsible for this generalization out of domain.
- footnote 8: Is there a motivating example for why solving batches of elections is important? That doesn't seem obvious
- Table 4 is not referenced in the text. Consider doing that
- Table 4 would be much more compelling if it considered how often all the rules being mimicked found the same winner. Are the networks learning the same thing when mimicking each rule or something different each time?
- there are a number of very relevant papers that have used neural networks to act as voting rules that you should strongly consider referencing, the mains ones are listed below


Typos:
- end of Sec 1: "in detail the proposed the training procedure"
- 3.2.1 line 3: "in the case of [a] fully-connected graph structure[s]" (should add one of [a] or [s])
- Table 3 caption: "and [are] mostly on par with,"
- 4.2, 2nd last paragraph: "The neural network models [are] better at"


References to add:

Xia, Lirong. "Designing social choice mechanisms using machine learning." Proceedings of the 2013 international conference on Autonomous agents and multi-agent systems. 2013.

- outlined the general approach most more recent papers have taken to use neural nets as voting rules

Armstrong, Ben, and Kate Larson. "Machine Learning to Strengthen Democracy." NeurIPS Joint Workshop on AI for Social Good. 2019.

- describes an approach to generate novel voting rules based on desirable axioms with neural networks

Burka, Dávid, et al. Voting: A machine learning approach. No. 145. KIT Working Paper Series in Economics, 2020.

- similar to (Kujawska et al. 2020), describes an approach to mimicking voting rules with neural MLPs

Firebanks-Quevedo, Daniel. "Machine Learning? In MY Election? It's More Likely Than You Think: Voting Rules via Neural Networks." 2020.

- similar to (Armstrong and Larson, 2019), generates voting rules automatically based on axioms and other desirable conditions

**Time Spent Reviewing:**

5

---

> ### Author Response · Authors · 2021-08-10
> **Thank you for your review**
>
> Thank you for your review - we find it very encouraging that you found that our results are likely to be built upon by other researchers!
>
> We’ll directly incorporate some of your recommendations in our revision. We’ll also expand our related works section to add the references you listed (thanks!). Here are our responses to some of your questions/considerations:
> * __Alpha:__ $\alpha$’s are the coefficients in the dirichlet distribution. In our experiments, the $\alpha_0$ values we used for “polarized” and “indecisive” preference distributions are 0.5 and 2.0 respectively. These are simply example values that fall within the required intervals ($(0, 1)$ for polarized, and $(1,\infty)$ for indecisive). One can certainly choose to experiment with other $\alpha_0$ values if needed. We will make sure to include this information in the revised version of the paper.
> * __Distribution of voters and candidates:__ We sampled the voter and candidate numbers uniformly within their corresponding intervals. Our networks do support a variable number of voters and candidates. While there is no strict upper bound on the number of voters, the candidate number is bounded above by the maximum number of candidates observed during training. We will add these clarifications to the paper.
> * __Training with two candidates:__ We agree that elections with 2 candidates are a relatively trivial case. We include this case for generality (in practice 2-candidate elections exist).
> * __How does training with variable number of voters affect generalization?__ This is a very interesting question! We trained the set transformer model to mimic Borda on only 99-voter elections. You can see how well this generalizes to other voter numbers.
>
> | Testing voter num. | 2-49 | 50-99 | 100-149 | 150-199 |
> |--------------------|------|-------|---------|---------|
> | Mimicking accuracy | 0.80 | 0.95  | 0.94    | 0.93    |
>
> These results indicate that training on a diverse input distribution is helpful for generalization.
>
> * __Batches:__ Batching could be useful if we need to compute the results of many elections frequently. For example, batching can be useful for voting systems that are part of recommended systems running on servers that are queried with elections very frequently. We’ll clarify this in the paper.
> * __How often the rules find the same winner?__
> Thank you for asking this question. We tested how well Plurality, Borda, Copeland and Maximin predict Borda and Kemeny winners on the uniform electorate. You can find the corresponding table below:
>
> |        | Plurality | Borda | Copeland | Maximin |
> |--------|-----------|-------|----------|---------|
> | Borda  | 0.30      | 1.0   | 0.74     | 0.65    |
> | Kemeny | 0.51      | 0.75  | 0.83     | 0.91    |
>
>   This shows that the elections we used to train the networks are nontrivial in that the aforementioned voting rules disagree on the winners to a considerable degree.

---

### Author Response · Authors · 2021-08-10
**General Comments**

We thank all the reviewers for their thoughtful and constructive reviews.

We find it encouraging that the reviewers found the performance of PIN architectures on voting rule mimicking and social welfare maximization tasks to be strong. We believe, as Reviewer 9xsn stated, that our results are in a position to be built upon by other researchers and would be a meaningful contribution to the NeurIPS community.

To address a common consideration in the reviews, we wanted to emphasize the significance of our voting rule mimicking experiments. Demonstrating strong performance on mimicking experiments is necessary (though not sufficient) for arguing that PIN architectures are well-suited for learning to represent voting rules. Moreover, the fact that the learned rules can generalize zero-shot to 1) an unseen number of voters 2) real world datasets 3) other synthetic distributions(*) suggest that  PIN architectures do indeed approximate the voting rules they are taught to mimic at an algorithmic level, instead of memorizing distinct mappings for particular voter counts or distributions.

(*) As requested by Reviewer sGWw, we’ve tested how well PIN architectures trained on elections sampled from uniform voter utility distributions zero-shot generalize to other synthetic distributions, such as polarized and indecisive (definitions in Section 4 of the paper). Our results indicate that the performance remains virtually the same (i.e. within ± 1 percentage point).

---

### Decision · Program_Chairs · 2021-09-27

**Decision:**

Accept (Poster)

**Comment:**

Majority of the reviewers are excited about the novel applications of ML techniques to social choice and were impressed by the good performance of the solution proposed in this paper. The paper is well-written and the work is solid. Some concerns were raised about the significance and motivation of the actual technical problem solved in this paper (i.e., learning social welfare maximizers) and technical depth in ML.

The response was generally effective and clarified some points raised by the reviewers. After the discussions, reviewers' opinions did not change much (except the score of one reviewer was raised from 6 to 7). The overall sentiment remained positive.

After all, the novelty and potential to stimulate future work and discussions outweigh the cons, which is the main reason behind the recommendation.